# Occurrence and Characteristics of *Staphylococcus aureus* Isolated from Dairy Products

**DOI:** 10.3390/molecules27144649

**Published:** 2022-07-21

**Authors:** Ewa Szczuka, Karolina Porada, Maria Wesołowska, Bogusława Łęska

**Affiliations:** 1Department of Microbiology, Institute of Experimental Biology, Faculty of Biology, Adam Mickiewicz University, Poznań, Uniwersytetu Poznańskiego 6, 61-614 Poznań, Poland; karolina.porada@onet.pl (K.P.); maria.wesolowska@amu.edu.pl (M.W.); 2Faculty of Chemistry, Adam Mickiewicz University, Poznań, Uniwersytetu Poznańskiego 8, 61-614 Poznań, Poland; bogunial@amu.edu.pl

**Keywords:** food safety, foodborne pathogens, *Staphylococcus aureus*, enterotoxin, antibiotic resistance, public health

## Abstract

Food, particularly milk and cheese, may be a reservoir of multi-drug resistant *Staphylococcus aureus* strains, which can be considered an important issue in terms of food safety. Furthermore, foods of animal origin can be a cause of staphylococcal food poisoning via the production of heat-stable enterotoxins (SE). For this reason, we investigated the prevalence of and characterized *Staphylococcus aureus* strains isolated from milk and fresh soft cheese obtained from farms located in Wielkopolskie and Zachodniopomorskie Provinces in Poland. Overall, 92% of *S. aureus* isolates were positive for at least one of the 18 enterotoxin genes identified, and 26% of the strains harbored 5 to 8 enterotoxin genes. Moreover, the *S. aureus* strains contained genes conferring resistance to antibiotics that are critically important in both human and veterinary medicine, i.e., β-lactams (*mecA*), aminoglycosides (*aac*(*6*′)/*aph*(*2*″), *aph*(*3*′)*-IIIa*, *ant*(*4*′)*-Ia*) and MLS_B_ (*erm*(*A*), *msr*(*A*), *lun*(*A*)). The antimicrobial susceptibility of *S. aureus* to 16 antibiotics representing 11 different categories showed that 74% of the strains were resistant to at least 1 antibiotic. Moreover, 28% of the strains showed multidrug resistance; in particular, two methicillin-resistant *S. aureus* strains (MRSA) exhibited significant antibiotic resistance. In summary, our results show that dairy products are contaminated by *S. aureus* strains carrying genes encoding a variety of enterotoxins as well genes conferring resistance to antibiotics. Both MRSA strains and MSSA isolates showing multidrug resistance were present in foods of animal origin.

## 1. Introduction

*Staphylococcus aureus* is a well-known opportunistic pathogen widely distributed in a broad range of hosts, including humans and animals. Foods from animal origin, such as milk and cheese, are susceptible to contamination by these bacteria [1]. *Staphylococcus aureus* have the ability to cause staphylococcal food poisoning (SFP) via the production of heat–stable enterotoxins (SE) [2,3]. It should be emphasized that staphylococcal food poisoning (SFP), caused mainly by *S. aureus* and very occasionally by other *Staphylococcus* species, is one of the most common food-borne diseases worldwide [4]. *S. aureus* can carry multiple *se* genes and produce different SEs and SE-like toxins. The SEs (SEA to SEE; SEG to SEI; SEK; SEM to SET) and SE-like toxins (SElJ; SElL; SElU to SElZ) were reported to be associated with food borne disease outbreaks [5]. Enterotoxin function as superantigens, i.e., they have the ability to stimulate large populations of T cells that cause unregulated activation of the immune response. If this stimulation is sustained, massive amounts of cytokines are produced, leading to a variety of acute toxic shock symptoms. These symptoms occur within hours after consumption of contaminated food and patients experience vomiting, diarrhea and abdominal pain. The illness is self-limited and usually resolves within 12–24 h [6].

The wide use of antibiotics in veterinary medicine has led to the emergence of multidrug-resistant strains (MDR), especially methicillin-resistant *S. aureus* (MRSA). Methicillin-resistant staphylococci (MRS) have an additional penicillin binding protein (PBP2a, encoded by either *mecA* or *mecC*), which confers resistance to beta-lactam antibiotics. Moreover, *S. aureus* has the ability to acquire resistance to antibiotics easily, including those commonly used in medicine, i.e., aminoglycosides as well as macrolides, lincosamides and streptogramin B (MLS_B_). Importantly, the MLSB antibiotics are the preferred alternative to penicillins and cephalosporins in the treatment of staphylococcal infection. Moreover, erythromycin and clindamycin are recommended as second-line drugs for patients with β-lactam allergy. Resistance to MLS_B_ is generally based on three mechanisms: ribosomal target modification mediated by *erm* genes, active efflux of antibiotics mediated by *msr*(*A*) and enzymatic drug inactivation mediated by *lnu*(*A*) [7]. Resistance to aminoglycosides is commonly related to drug inactivation by cellular aminoglycoside-modifying enzymes, especially the bifunctional enzyme AAC(6′)/APH(2″), encoded by *aac*(*6*′)/*aph*(*2*″), the APH(3′)-III enzyme encoded by *aph*(*3*′)*-IIIa* and the ANT(4′)-I enzyme encoded by *ant*(*4*′)*-Ia*. It is well known that the MRSA strains have been isolated from several food-producing animals, including pigs, cattle and chickens [5,8]. The probability of transferring these bacteria to milk and cheeses, particularly those made from raw milk, is high [9,10]. The consumption of these products may lead to asymptomatic colonization of the intestinal tract by resistant food-borne bacteria. Antimicrobial therapy during this colonization period may lead to the development of a severe clinical disease because the resistant food-borne pathogens may prevail over the sensitive antagonistic gut microbiota [8].

Milk and cheese are naturally susceptible to contamination by *S. aureus*, which can multiply and produce enterotoxins. Most studies have focused on the detection of genes encoding classical enterotoxins, whereas knowledge about the prevalence of newer enterotoxins is limited. Therefore, this study aimed to estimate the presence of 18 genes coding for enterotoxins in *S. aureus*. We also evaluated the occurrence of genes encoding resistance to antibiotics that are often used in human medicine, i.e., β-lactams (*mecA*), aminoglycosides (*aac*(*6*′)/*aph*(*2*″), *aph*(*3*′)*-IIIa*, *ant*(*4*′)*-Ia*) and MLSB (*erm*(*A*), *erm*(*B*), *erm*(*C*), *lnu*(*A*) *msr*(*A*)). We intended to estimate whether raw milk and cheese are sources of enterotoxigenic and multidrug-resistant *S. aureus* strains.

## 2. Result and Discussion

Although *S. aureus* has been well recognized for its ability to evoke staphylococcal food poisoning via the production of enterotoxins, only a limited number of reports have focused on the detection of newly described staphylococcal enterotoxins. In this study, all the *S. aureus* isolates were evaluated for the presence of 5 classical SE genes (*sea*, *seb*, *sec*, *sed*, *see*) and 13 newer SE genes (*seg*, *seh*, *sei*, *sej*, *sek*, *sel*, *sem*, *sen*, *seo*, *sep*, *seq*, *ser and seu*). They were found in 36 isolates (92%), which is a higher number than that found by Kou et al. [11]. However, in their study, Kou et al. tested only the five classical enterotoxins (*sea*, *seb*, *sec*, *see* and *sed*) in *S. aureus* isolates from retail raw milk from four regions in northern Xinjang, China. In our study, 18 enterotoxin genes (*sea*, *seb*, *sec*, *sed*, *see*, *seg*, *seh*, *sei*, *selj*, *sek*, *sel*, *sem*, *sen*, *seo*, *sep*, *seq*, *ser*, *seu*) and 26 different toxin gene profiles were detected among *S. aureus* isolates. Among them, the main profiles were *sec*, *seg*, *sel*, *sei*, *sem*, *sen*, *seo* and *seu* (10%, 4/39). Remarkably, 26% of the strains harbored five to eight enterotoxin genes (Table 1). The most frequent *se* gene detected was *seo*, followed by *sek* and *slu*. Various studies have reported different prevalence rates of the enterotoxin genes, which may be attributed to the type of food product and geographical distribution. For example, French studies have reported that *sea* genes were present in *S. aureus* from various food products most frequently [12]. Kou et al. [11] reported that the *see* gene was the most commonly detected SE in raw milk in China. In contrast, the *see* gene was not found in any of the *S. aureus* isolates analyzed in the present study.

Importantly, the multidrug resistance (MDR) phenotype (resistant to ≥3 subclasses of antimicrobial drugs) was observed in the *S. aureus* strains. The overall prevalence of MDR *S. aureus* isolates was 28%. All the MDR phenotypes included resistance to penicillin, i.e., an antibiotic that is widely used in veterinary medicine. Two isolates (5.1%) were verified as MRSA by the cefoxitin disc diffusion test and were positive for the *mecA* gene. Similarly, a low prevalence of MRSA in milk samples from dairy cows was reported in Germany [13]. MRSA isolates were not found in bulk tank milk collected from 91 different farms in Denmark [14]. In studies conducted in Croatia, Cvetnić et al. reported that methicillin resistance was detected in 10 (4.2%) *S. aureus* strains isolated from milk samples of cows suffering from subclinical mastitis [3]. A study from China reported a high MRSA prevalence (51.6 %) in raw milk samples [11]. In our study, the MRSA isolate MPU Sa 103 showed resistance to β-lactams, aminoglycosides and macrolides, while the other MRSA isolate, MPU Sa, 118 showed resistance to lincosamides, fluoroquinolones, tetracyclines, rifampicin and trimethoprim/sulfamethoxazole in addition to the antibiotic classes to which MPU Sa 103 was resistant, as shown in Table 2. As mentioned above, our results showed relatively high resistance to the β-lactams penicillin (59%) and ampicillin (36%). The observed high levels of resistance to penicillins were in consonance with previously published data demonstrating β-lactam resistance in staphylococci isolated from food products [15,16,17]. Approximately one-third of the strains displayed resistance to the aminoglycosides gentamycin (31%) and tobramycin (26%). Seven gentamycin-resistant and tobramycin-resistant isolates were found to carry the aminocyclitol-6′-acetyltransferase-aminocyclitol-2″-phosphotransferase (*aac*(*6*′)/*aph*(*2*″)) gene. The minocyclitol-3′-phosphotransferase (*aph*(*3*′)*-IIIa*) gene was detected in 28% of the strains, whereas the aminocyclitol-4′-adenylyltransferase (*ant*(*4*′)*-Ia*) gene was found in 10% of strains. Noteworthy, two pan-susceptible isolates (susceptible to all antibiotics) carried the *ant*(*4*′)*-Ia* gene. One isolate contained two AME genes, *aph*(*3*′)*-IIIa* and *ant*(*4*′)*-Ia*. Coexisting *aac*(*6*′)/*aph* (*2*″) and *aph*(*3*′)*-IIIa*) genes were detected in six strains. The examined strains harbored genes conferring resistance to MLS_B_. However, the prevalence of these resistance genes was generally low. Four strains carried the *msr*(*A*) gene, three isolates had the *erm*(*B*) gene and a single isolate had the *lnu*(*A*) gene. None of the isolates harbored *erm*(*A*) and *erm*(*C*) genes. The presence of macrolide resistance genes has been reported previously for *S. aureus* isolates from samples of milk obtained from dairy cows [18]. Erythromycin resistance was found in seven (18%) strains, whereas resistance to clindamycin was found in five strains (13%). These studies showed that the erythromycin resistance rate in *S. aureus* in Poland was lower than in other countries. Erythromycin resistance was detected in 74% of *S. aureus* from raw aquatic food in China, 46% of *S. aureus* from livestock animals in Italy and 47% of *S. aureus* from raw meat in the Czech Republic [15,19,20]. Additionally, a lower rate of tetracycline resistance (18%) was observed compared with previous reports from the Czech Republic, in which almost all MRSA strains isolated from samples of cow, sheep and goat milk showed resistance to tetracycline [21]. In our investigations, we detected five (13%) rifampicin-resistant strains and only one (2.5%) trimethoprim/sulfamethoxazole-resistant strain. Resistance to ciprofloxacin, levofloxacin and moxifloxacin was revealed in 23, 20 and 15% of the strains, respectively. Fortunately, none of the 39 isolates showed resistance to the critically important antimicrobial agent vancomycin. It is important to highlight a study conducted by Bhattacharyya et al. [22] that demonstrated the occurrence of vancomycin-resistant *S. aureus* isolated from bovine and caprine milk. In this study, no resistance to tigecycline and chloramphenicol was recorded. All the 39 isolates exhibited susceptibilities to vancomycin, tigecycline and chloramphenicol. Of the 39 *S. aureus* isolates, 10 were susceptible to all antibiotics, 7 were resistant to at least one antibiotic, and the other 22 isolates were resistant to two or more antibiotics. Interestingly, studies from Norway indicated that all S. aureus strains isolated from milk and cheese were sensitive to the 12 antibiotics tested [9]. In contrast, 80.6% of *S. aureus* strains isolated from milk in China were resistant to at least one antibiotic, which is close to our findings [11].

Taken together, our results show that milk and cheese are contaminated by *S. aureus* strains carrying genes encoding a variety of enterotoxins as well genes conferring resistance to antibiotics. Moreover, our findings show that the frequency of multidrug-resistant *S. aureus* strains in food is alarming and may represent a public health problem.

## 3. Material and Methods

### 3.1. Collection of Samples

Milk and cheese from farms located in the Wielkopolskie and Zachodniopomorskie Provinces were subjected to the analysis. The farms were semi-subsistence farms with three to seven dairy cows. The bulk milk and cheese made from raw milk was collected during one year. In total, 62 samples were analyzed. The samples were collected in sterile containers and transported immediately to the microbiological laboratory. The milk samples were centrifuged and a loop of sediment (10 μL) was streaked onto on Baird–Parker agar (Oxoid Ltd., Basingstoke, Hampshire, UK). Samples of curd were diluted 1:10 in sterile Ringer’s solution and homogenized in blender, and 100 μL of samples were placed on Baird–Parker agar (Oxoid Ltd.). Plates were incubated at 37 °C and examined after 24–48 h for bacterial growth.

### 3.2. Bacterial Isolation and Identification

Staphylococci colonies were subcultivated on brain–heart infusion agar. The strains were tested for bound coagulase by slide coagulase test on sterile microscope slides and free coagulase by tube coagulase test with rabbit plasma (Oxoid). Presumptive *S. aureus* were confirmed by API STAPH test strips (bioMerieux, Marcyl’Etoile, France). A total of 39 *S. aureus* strains were identified. Of these, 9 strains were isolated from cheese and 30 from milk. The bacterial strains used in the study were stored in brain–heart infusion agar (BHI, Oxoid) with 20% glycerol at −80 °C.

### 3.3. Susceptibility Testing

The antimicrobial susceptibility of the staphylococci was tested by using the agar disc diffusion method according to the CLSI standards for the following antimicrobial agents (µg/disc): ampicillin (25), cefoxitin (30), ciprofloxacin (5), clindamycin (2), chloramphenicol (30), erythromycin (15), gentamicin (10), levofloxacin (5), moxifloxacin (5), penicillin G (10), rifampicin (5), tetracycline (30), tigecycline (1,5), tobramycin (10), trimethoprim/sulfamethoxazole (1.25 + 23.75) and vancomycin (30).

### 3.4. Preparation of Total DNA for PCR and Detection of SE Genes and Antibiotic Resistance Genes

The total DNA was isolated and purified using the Genomic Mini DNA kit (A&A Biotechnology, Gdynia, Poland). The presence of se genes (*sea*, *seb*, *sec*, *sed*, *see*, *seg*, *seh*, *sei*, *selj*, *sek*, *sel*, *sem*, *sen*, *seo*, *sep*, *seq*, *ser*, *selu*) and antibiotic resistance genes *mecA*, *aac*(*60*)/*aph*(*200*), *aph*(*30*)*-IIIa*, *ant*(*40*)*-Ia*, *erm*(*A*), *erm*(*B*), *erm*(*C*), *msr*(*A*), *lun*(*A*) was assessed using PCR assays as described previously (Table 3) [23,24,25,26].

## 4. Conclusions

Contamination of food by pathogenic bacteria is a fundamental issue in the area of food safety since it can compromise human health. Our results show that raw milk and cheese are contaminated by *S. aureus.* These strains carry genes encoding a variety of enterotoxins that are responsible for staphylococcal food poisoning. Furthermore, *S. aureus* strains contain genes conferring resistance to antibiotics (i.e., β-lactams, aminoglycosides and MLS_B_), which are critically important in both human and veterinary medicine. Thus, the consumption of unpasteurized milk and other dairy products may lead to asymptomatic colonization of the intestinal tract by resistant food-borne bacteria.

## Figures and Tables

**Table 1 molecules-27-04649-t001:** Toxin gene profiles.

Toxin Gene Profiles	Number of Isolates	Origin of Isolates (No) *
*sec*, *seg*, *sel*, *sei*, *sem*, *sen*, *seo*, *seu*	4	milk (3) cheese (1)
*seb*, *seh*, *sek*, *seu*, *seq*	2	milk (1) cheese (1)
*sek*, *seu*	2	milk (1) cheese (1)
*seb*, *seh*, *sek*, *seq*	2	milk (1) cheese (1)
*sec*, *seg*, *sei*, *sel*, *sen*, *seo*	2	milk (1) cheese (1)
*sea*, *seh*	2	milk
*sek*, *seo*	2	milk
*seo*	2	milk (1) cheese (1)
*sec*, *sei*, *sem*, *seo*, *sen*, *sel*, *seu*, *ser*	1	milk
*seh*, *seg*, *sei*, *sek*, *seo*, *ser*, *sen*, *seu*	1	milk
*sec*, *seg*, *sei*, *sel*, *sem*, *sen*, *seu*	1	milk
*sec*, *sed*, *sei*, *seg*, *selj*, *seo*, *sem*	1	milk
*seg*, *sei*, *sem*, *seo*, *ser*, *sen*, *seu*	1	milk
*seb*, *sek*, *ser*, *seu*, *seq*	1	milk
*sed*, *seg*, *sei*, *seo*, *sem*	1	milk
*seg*, *sen*, *seo*, *seu*	1	milk
*sec*, *seg*, *sei*, *seo*	1	chesse
*seb*, *seg*, *seh*, *sei*	1	milk
*seg*, *ser*, *sen*, *seu*	1	milk
*sea*, *seo*, *sek*	1	cheese
*sea*, *sek*, *seq*	1	milk
*sep*, *sek*, *seo*	1	cheese
*seb*, *sek*, *seq*	1	milk
*seo*, *sek*	1	milk
*sek*, *seq*	1	milk
*sep*	1	milk

(No) * Number indicating the origin of *Staphylococcus aureus* strains.

**Table 2 molecules-27-04649-t002:** Antimicrobial resistance and prevalence of antibiotic resistance genes among *Staphylococcus aureus*.

Antibiotic Profiles	Number of Antibiotics	Total Number of Isolates	Presence of *mecA* Gene (No.) *	Presence of *aac*(*6*′)/ *Aph*(*2*″) Gene (No.)	Presence of *ap*(*3*′)*-IIIa* Gene (No.)	Presence of *Ant*(*4*′)*-Ia* Gene (No.)	Presence of *msr*(*A*) Gene (No.)	Presence of *erm*(*A*) Gene (No.)	Presence of *erm*(*B*) Gene (No.)	Presence of *erm*(*C*) Gene (No.)	Presence of *lnu*(*A*) Gene (No.)
P	1	4	0	0	0	0	1	0	1	0	0
CN	1	1	0	0	0	1	0	0	0	0	0
TE	1	1	0	0	1	0	0	0	1	0	0
RD	1	1	0	0	0	0	0	0	0	0	0
P, E	2	1	0	0	0	0	0	0	0	0	0
AMP, P	2	5	0	0	1	1	0	0	0	0	0
P, E, DA	3	1	0	0	0	0	0	0	0	0	1
CN, TOB, DA	3	1	0	1	0	0	0	0	0	0	0
CIP, LEV, MXF	3	1	0	0	0	0	0	0	0	0	0
AMP, P, E	3	1	0	0	0	0	0	0	0	0	0
AMP, P, CN, TOB	4	1	0	0	0	0	0	0	0	0	0
CIP, LEV, MXF, CN	4	1	0	0	1	0	1	0	0	0	0
P, CIP, LEV, E, RD	5	1	0	0	1	0	1	0	0	0	0
AMP, FOX, P, TOB, E	5	1	1	0	0	0	0	0	0	0	0
P, CIP, LEV, CN, TOB	5	1	0	1	1	0	0	0	0	0	0
P, CIP, LEV, MXF, CN, TOB, TE	7	1	0	1	1	0	0	0	0	0	0
AMP, P, CIP, LEV, MXF, CN, E	7	1	0	0	1	0	1	0	0	0	0
AMP, P, CN, TOB, DA, TE, RD	7	2	0	2	2	0	0	0	1	0	0
AMP, P, CIP, LEV, MXF, CN, TOB, TE	8	2	0	2	2	0	0	0	0	0	0
AMP, FOX, P, CIP, CN, TOB, E, DA, TE, RD, SXT	11	1	1	0	0	0	0	0	0	0	0

* Number indicating the prevalence of antibiotic resistance genes in *Staphylococcus aureus* strains. AMP, ampicillin; CIP, ciprofloxacin; CN, gentamycin; DA, clindamycin; E, erythromycin; FOX, cefoxitin; LEV, levofloxacin; MXF, moxifloxacin; P, penicillin; RA, rifampicin; SXT, sulfamethoxazole/trimethoprim; TE, tetracycline; TOB, tobramycin.

**Table 3 molecules-27-04649-t003:** Primers used in this study.

Gene	Primer	Oligonucleotide Sequence (5′ to 3′)	Reference
*sea*	SEA-F SEA-R	CAGCATACTATATTGTTTAAAGGC CCTCTGAACCTTCCCATC	[23]
*seb*	SEB-F SEB-R	GTATGGTGGTGTAACTGAGCA TCAATCTTCACATCTTTAGAATCA	[23]
*sec*	SEC-F SEC-R	CTCAAGAACTAGACATAAAAGCTAGG TCAAAATCGGATTAACATTATCC	[23]
*sed*	SED-F SED-R	CTAGTTTGGTAATATCTCCTTTAAACG TTAATGCTATATCTTATAGGGTAAACATC	[23]
*see*	SEE-F SEE-R	CAGTACCTATAGATAAAGTTAAAACAAGC TAACTTACCGTGGACCCTTC	[23]
*seg*	SEG-F SEG-R	AAGTAGACATTTTTGGCGTTCC AGAACCATCAAACTCGTATAGC	[23]
*seh*	SEH-F SEH-R	GTCTATATGGAGGTACAACACT GACCTTTACTTATTTCGCTGTC	[23]
*sei*	SEI-F SEI-R	GGTGATATTGGTGTAGGTAAC ATCCATATTCTTTGCCTTTACCAG	[23]
*sej*	SEJ-F SEJ-R	ATAGCATCAGAACTGTTGTTCCG CTTTCTGAATTTTACCACCAAAGG	[23]
*sek*	SEK-F SEK-R	TAGGTGTCTCTAATAATGCCA TAGATATTCGTTAGTAGCTG	[23]
*sel*	SEL-F SEL-R	TAACGGCGATGTAGGTCCAGG CATCTATTTCTTGTGCGGTAAC	[23]
*sem*	SEM-F SEM-R	GGATAATTCGACAGTAACAG TCCTGCATTAAATCCAGAAC	[23]
*sen*	SEN-F SEN-R	CATCATGCTTATACGGAGGAG CCCACTGAACCTTTTACGTT	[23]
*seo*	SEO-F SEO-R	TGTGTAAGAAGTCAAGTGTAG TCTTTAGAAATCGCTGATGA	[23]
*sep*	SEP-F SEP-R	TGATTTATTAGTAGACCTTGG ATAACCAACCGAATCACCAG	[23]
*seq*	SEQ-F SEQ-R	TCAAGGAGTTAGTTCTGGAAATT GCTTACCATTGACCCAGAGA	[23]
*ser*	SER-F SER-R	GGATAAAGCGGTAATAGCAG GTATTCCAAACACATCTAAC	[23]
*sel*	SEU-F SEU-R	ATCAGAAACAAACATTAAAGCCCA TGACCATTTCCTTCGATAAACTTTAT	[23]
*mecA*	MecA-F MecA-R	GTGAAGATATACCAAGTGATT ATGCGCTATAGATTGAAAGGAT	[24]
*aac*(*6*′)/*aph*(*2*″)	AAC(6′)/APH(2″)-F AAC(6′)/APH(2″)-R	GAAGTACGCAGAAGAGA ACA TGG CAA GCT CTA	[25]
*aph*(*3*′)*-IIIa*	APH(3′)-IIIa-F APH(3v)-IIIa-R	AAATACCGCTGCGTA CATACTCTTCCGAGCAA	[25]
*ant*(*4*′)*-Ia*	ANT(4′)-Ia-F ANT(4′)-Ia-R	AATCGGTAGAAGCCCAA GCACCTGCCATT GCTA	[25]
*erm*(*A*)	ermA-F ermA-R	TCTAAAAAGCATGTAAAAGAAA CGATACTTTTTGTAGTCCTTC	[26]
*erm*(*B*)	ermB-F ermB-R	CCGTTTACGAAATTGGAACAGGTAAAGGGC GAATCGAGACTTGAGTGTGC	[26]
*erm*(*C*)	ermC-F ermC-R	GCTAATATTGTTTAAATCGTCAATTCC GGATCAGGAAAAGGACATTTTAC	[26]
*msr*(*A*)	msrA-F msrA-R	TGCTGACACAATTTGGGAT GAGCAGCCTTCTCAACC	[26]
*lnu*(*A*)	linA-F linA-R	GGTGGCTGGGGGGTAGATGTATTAACTGG GCTTCTTTTGAAATACATGGTATTTTTCGA	[26]

## Data Availability

Not applicable.

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
