# Peer review of "Occurrence and Characteristics of Staphylococcus aureus Isolated from Dairy Products"

_molecules, 2022, doi:10.3390/molecules27144649_

Round 1

Reviewer 1 Report

This manuscript investigated the prevalence and characterized Staphylococcus aureus strains isolated from milk and cheese in Poland and found more than 90% of S. aureus isolates were positive for the 17 enterotoxin genes identified and 26% of the strains harbored five to eight enterotoxin genes. This might be a manuscript investigated the enterotoxins genes, however, there are some parts needed to be modified.

1. Authors said Staphylococcus aureus strains isolated from milk and cheese in Poland, and 92% of S. aureus isolates were positive for at least one of the 17 enterotoxin genes. However, these is no information about the sample size and total Staphylococcus aureus strain numbers.

2. Most studies focused on the detection of genes encoding classical enterotoxins, whereas the knowledge about the prevalence of newer enterotoxins is limited.

What are the newer enterotoxins? Is there any introduction?

3. The research on Staphylococcus aureus has been in-depth, so what is the highlights for this research? New in Poland?

4. The discussion is too less, and lots of results have only been introduced without any discussion. I donot think no research has been performed before.

5. English needs to be improved. For example, line 75, weather should be whether. MLSB and MLSB, which is correct?

Author Response

I hereby resubmit our manuscript entitled: "Occurrence and characteristics of Staphylococcus aureus isolated from dairy products".

We would extremely grateful to the Reviewers for advising us of the opportunity to have our article corrected and improved, a service of which we have readily taken advantage.

All corrections have been made according to the reviewers’ suggestions.

Reviewer # 1

Following the Reviewer’s suggestions, the following have been improved:

  1. We have added information about the sample size and total Staphylococcus aureus strain numbers.

In total, 62 samples were analysed. (page 8, lines 5-6).

A total of 39 S. aureus strains were identified. (page 8, lines 16-17).

  1. As the Reviewer suggested, we have put into the Result and Discussion (page 3, lines 30- 32) the following sentence: In this study, all the S. aureus isolates were evaluated for the presence of five classical SE genes (sea, seb, sec, sed, see) and 13 newer SE genes (seg, seh, sei, sej, sek, sel, sem, sen, seo, sep, seq, ser and seu).

In earlier studies, we found the term "newer SE genes". However other authors used the term “newly described staphylococcal SE”, or “new types of SEs” to distinguish them from classic enterotoxins.

  1. Our results showed that dairy products are contaminated by S. aureus strains carrying genes encoding a variety of enterotoxins. In addition to this, antibiotic susceptibility evaluation highlighted high levels β- lactams resistance among staphylococci that may indicate an selection of resistant bacteria due to the use of antibiotics in farms. Additionally, we demonstrated the occurrence of methicillin-resistant S. aureus (MRSA) in foods of animal origin.

In Poland, extensive research was carried out on the occurrence of S. aureus on pig farms located in 14 provinces. Seventy-nine isolates were isolated, including methicillin-resistant strains. (Mroczkowa et.al, 2017; PLOS ONE  DOI:10.1371/journal.pone.0170745). Fijałkowski et.al (2013) analysed staphylococci isolated from ready-to-eat meat products, including pork ham, chicken cold cuts, pork sausage, salami and pork luncheon meat. However, all isolated staphylococci were identified as Staphylococcus equorum, S. vitulinus, S. carnosus, S. succinus , S. xylosus, S. saprophyticus, S. warneri, S. haemolyticus and S. pasteuri (https://doi.org/10.1016/j.ijfoodmicro.2016.09.001). Earlier, Nawroted et.al (2010) analysed the presence of enterotoxin genes (sea- see) among staphylococci isolated from cows. Only two S. aureus strains were included in this study.

  1. We are grateful to the Reviewer for highlighting the fact that we had not discussed all results. Consequently, the following sentence has been added:

The observed high levels of resistance to penicillins were in consonance with previously published data demonstrating β-lactam resistance of staphylococci isolated from food products [16,17,18].(page 4, lines 26- 28)

The presence of macrolide resistance genes has been reported previously for S. aureus isolates from samples of milk obtained from dairy cows [19].(page 5, lines 4- 5).

These studies showed that the erythromycin resistance rate in S. aureus in Poland was lower than in other countries. Erythromycin resistance was detected in 74% of S. aureus from raw aquatic food in China, 46% of S. aureus from livestock animals in Italy, and 47% of S. aureus from raw meat in the Czech Republic [16, 20, 21]. Also, a lower rate of tetracycline resistance (18%) was observed, compared with previous reports from Czech Republic, in which almost all MRSA strains isolated from samples of cow, sheep and goat milk showed resistance to tetracycline [22]. (page 5, lines 7- 12). 

Fortunately, none of the 39 isolates showed resistance to the critically important antimicrobial agent, i.e. vancomycin. It is important to highlight a study conducted by Bhattacharyya et al. [23] demonstrated the occurrence of vancomycin - resistant S. aureus isolated from bovine and caprine milk. (page 5, lines 15- 18). 

Interestingly, studies from Norway indicated that all S. aureus strains isolated from milk and cheese were sensitive to the 12 antibiotics tested [9]. In contrast, 80.6% of S. aureus strains isolated from milk in China were resistant to at least one antibiotic, which is close to our findings [12]. (page 5, lines 21- 24). 

  1. We used MLSB throughout the article. The text has been checked by a native speaker.

Reviewer 2 Report

The reviewed paper is very important for both veterinary and medicine. The authors presented that milk and fresh soft cheese can be sources of one of the most often human pathogen, Staphylococcus aureus. In the study, 92% of S. aureus isolates contained enterotoxin genes and 74% of the strains were resistant to at least one antibiotic. The authors showed also that 28% of strains are multidrug-resistant, including MRSA. These results indicate that the widespread use of antibiotics in animals leads to the development of multidrug resistance in bacteria. At the same time, animal products can be potentially dangerous to humans and any possible infection can be difficult to treat due to drug resistance. The methodology is proper and the results are well presented. I would like suggest 2 corrections:

1. The authors can write, which strains were isolated from milk and which from cheese. Maybe will be interesting differences?

2. In point 3.4., please add a table with used primers. The authors gave references, but I think, showing of primers sequences will allow other researchers to directly cite this publication.

Author Response

Dear Reviewer, 

I hereby resubmit our manuscript entitled: "Occurrence and characteristics of Staphylococcus aureus isolated from dairy products".

We would extremely grateful to the Reviewers for advising us of the opportunity to have our article corrected and improved, a service of which we have readily taken advantage.

Following the Reviewer’s suggestions, the following have been improved:

  1. We have added information about origin of strains ( table 1). Unfortunately, we haven’t find interesting differences.
  2. We added a table 3 with used primers.

Your sincerely,

Ewa Szczuka

Round 2

Reviewer 1 Report

The comments have been well addressed. The manuscript can be accept for publication.